# Molecular Characterization, Expression, and Regulatory Signal Pathway Analysis of Inflammasome Component Apoptosis-Associated Speck-like Protein Containing a CARD Domain (ASC) in Large Yellow Croaker (*Larimichthys crocea*)

**DOI:** 10.3390/ijms24032175

**Published:** 2023-01-22

**Authors:** Xin Tang, Xiande Liu, Zhiyong Wang, Meiling Chen, Dongling Zhang

**Affiliations:** 1Key Laboratory of Healthy Mariculture for the East China Sea, Ministry of Agriculture and Rural Affairs, Jimei University, Xiamen 361021, China; 2Laboratory for Marine Fisheries Science and Food Production Processes, Qingdao National Laboratory for Marine Science and Technology, Qingdao 266032, China

**Keywords:** ASC, inflammasome, gene expression, transcriptome, large yellow croaker

## Abstract

ASC (apoptosis-associated speck-like protein containing a caspase recruitment domain (CARD)) is the only adaptor involved in the formation of multiple types of inflammasomes. Accumulating evidence demonstrates that ASC plays a critical role in the protection of the host against pathogen infection. In this study, we identified an *ASC* gene in the large yellow croaker (*Larimichthys crocea*), namely *LcASC*, and then investigated the expression characteristics and related signal pathways. On one hand, LcASC has several conserved protein modules, i.e., an N-terminal PYD region, a C-terminal CARD region, and twelve α-helix structures. On the other hand, it has a high variable linker between PYD and CARD domains. Moreover, *LcASC* has varying degrees of expression in different tissues, among which the highest expression is observed in the spleen followed by the gills and skin. It also shows induced expressions in the head kidney, liver, and spleen following immune stimulation, especially *Vibrio Parahaemolyticus* infection. Further subcellular localization analysis showed that LcASC formed a clear aggregated speck in the cytoplasm close to the nucleus. In addition, we found 46 DEGs in a comparative transcriptome analysis between the LcASC overexpression group and the control vector group. Notedly, the up-regulated gene *Fos* and down-regulated gene *DOK3* in LcASC overexpressed cells play important roles in the immune system. How ASC contacts these two genes needs to be clarified in upcoming studies. These findings collectively provide new insights into finfish ASC and its potential regulatory signaling pathway as well.

## 1. Introduction

Innate immunity is the first line of defense against infection. Inflammasome, an innate immune signaling platform, is a multi-protein complex comprised of a sensor protein, an adapter protein, and an effector protein [1]. It has been reported that pattern recognition receptors (PRRs), such as NLRP1/2/3/6//12 and AIM2 [2,3,4,5], can serve as sensor proteins recognizing pathogen-associated molecular patterns (PAMPs) (e.g., LPS and mitochondrial DNA) and danger-associated molecular patterns (DAMPs) (e.g., high mobility group box 1 (HMGB1) and ATP) [6,7,8]. Furthermore, pro-Caspase-1 can act as an effector protein and trigger the maturation and secretion of potent pro-inflammatory cytokines (IL-1β and IL-18), initiating the cascade reaction of an inflammatory response [5]. The apoptosis-associated speck-like protein CARD domain (ASC) is an adapter protein to bridge the upstream sensor proteins PRRs with downstream effector protein pro-Caspase-1.

Due to its central role in the inflammasome complex, ASC has been studied extensively in vertebrates. In mammals, ASC, also known as pycard or the target of methylation-induced silencing-1 (TMS-1), is a 22 kDa cytosolic protein composed of an N-terminal PYRIN domain (PYD) and a C-terminal caspase recruitment domain (CARD) [9]. In most mammals, the NLR family recognizes the ligands and binds to ASC and pro-Caspase1 via PYD–PYD and CARD–CARD interactions, thus forming the inflammasome to further induce inflammation and pyroptosis. In humans, the inflammasome is closely associated with various infectious diseases, nervous system diseases, metabolic diseases, and cardiovascular diseases [5]. In a mouse model, accumulating evidence strongly demonstrates that ASC plays a critical role in the protection of the host against pathogen infection [10,11,12,13,14,15,16,17,18,19]. *ASC*-deficient mice exhibited dramatic susceptibility to west Nile virus infection, and the expression levels of IFN-α, IgM and pro-inflammatory cytokines were significantly decreased [10]. *ASC*-/- mice were extremely susceptible to *Anaplasma phagocytophilum* compared to WT (wild type) mice due to the absence of IL-18 secretion and reduced IFN-γ levels in the peripheral blood [11]. In addition, the host defense in chronic infection with *Mycobacterium tuberculosis* depends on ASC, but not on NLRP3 or caspase 1 [12]. The induction of antigen-specific IgG antibodies to influenza vaccines is impaired in mice deficient in ASC, but not in NLRP3 or caspase 1 deficiency [13]. Similar phenotypes have also been observed with *Helicobacter pylori* [14], *Legionella pneumophila* [15], *Staphylococcus aureus* [16], *Listeria monocytogenes* [17], and *Candida albicans* [18]. In addition, ASC is a dual regulator of the NF-κB pathway, and it can either enhance or suppress NF-κB activity depending on various stimuli [20].

In teleost, ASC has been reported in several fish, including mandarin fish (*Siniperca chuatsi*) [21], Japanese flounder (*Paralichthys olivaceus*) [22], goldfish (*Crassius auratus*) [23], zebrafish (*Danio rerio*) [24], orange-spotted grouper (*Epinephelus coiodes*) [25], turbot (*Scophthalmus maximus*) [26], and Japanese medaka (*Oryzias latipes*) [27,28]. The results of these studies indicated that teleost ASCs were induced to different extents in response to bacterial infections, nigericin, ATP, and other pattern immunostimulants. ASC-knockout (KO) medaka demonstrated higher mortality than WT after being challenged with *Aeromonas hydrophila* [27]. However, the immune mechanism of ASC in commercial fish remains largely unknown. In particular, there is relatively little published information on its regulatory signal pathway.

The large yellow croaker (*Larimichthys crocea*), as a commercial marine fish, is the largest cultivated production mariculture fish in China. Net-cage cultivation is the main farming mode for fish. However, high-density net-cage cultivation usually causes the outbreak of many diseases, especially bacteria and parasites [29,30]. Zhang et al. reported that inflammasome components NLRP1, NLRP3, and ASC were significantly induced by a trivalent bacterial vaccine through transcriptome analysis [31]. However, how the inflammasome is involved in fish disease remains unclear. Hence, it is an urgent requirement to explore the ASC immune mechanism in the large yellow croaker and to further investigate the inflammasome’s role.

In the current study, the *ASC* gene in *L. crocea* (named LcASC) was identified and its tissue expression profile was detected. Meanwhile, the responses of *LcASC* to the infection of *Vibrio parahaemolyticus* and the stimulation of LPS and Poly I:C were fully investigated. Moreover, the subcellular localization of the protein was determined, and its regulatory signal pathway was preliminarily explored.

## 2. Results

### 2.1. Sequence Analysis of LcASC Gene

The nucleotide sequence of *LcASC* cDNA contains a 582 bp ORF encoding a predicted protein of 193 aa, with a molecular weight of 21.86 kDa and an isoelectric point of 5.89. Sequence alignment shows that LcASC shares 31.0%–84.86% overall sequence identities with those of other teleost species including *Collichthys lucidus*, *O. Latipes,* and *D. rerio*. The overall sequence identity between LcASC and human ASC is 31.0%. SMART analyses indicate that LcASC contains a conserved N-terminal PYD (6–86 aa) and a C-terminal CARD domain (104–192 aa) (Figure 1A). Multiple alignment analyses showed that these functional domains are conserved from fish to mammals while the link sequence between the PYD domain and the CARD domain is highly variable. Furthermore, both domains consist of 12 α-helix structures based on PSIPRED and SWISS-MODEL predictions (Figure 1A,B). In addition, some charged residues for mammalian PYD or CARD fibrillation were highly conserved in large yellow croaker ASC, including D12, E15, R37, R40, and D50 in the PYD domain and D130, Y143, R147, R157, and D188 in the CARD domain. In addition, LcASC possesses 16 phosphorylation sites required for inflammasome assembly [32].

To determine the evolutionary relationship between LcASC and other species’ ASCs, a phylogenetic tree was constructed using the neighbor-joining method based on multiple sequence alignment (Figure 1C). The amino acid sequences of LcASC and other ASCs from different vertebrates are collected. The LcASC first clusters with *C. lucidus* ASC, and then clusters with the same proteins from other teleosts, suggesting a close relationship between LcASC and homologues from teleosts. In addition, the cluster from fish ASC is separated from those of mammals and amphibians.

### 2.2. Tissue Distribution of LcASC Gene in Healthy Large Yellow Croaker

The tissue distribution of the *LcASC* gene in healthy large yellow croaker was determined by Real-Time Quantitative PCR (qPCR). The results showed that the *LcASC* gene was ubiquitously expressed in all detected tissues, with the highest level of expression in the spleen, followed by skin and gills, weakly in muscle, heart, head kidney, and intestine, and negligibly in blood, liver, stomach, brain, and kidney (Figure 2). Notably, the expression level of *LcASC* in the spleen, gills, and skin was more than 12,372-fold, 902.25-fold, and 429.29-fold higher than that in the kidney, respectively.

### 2.3. LcASC Gene Expression in Response to Immune Stimulation

To investigate the expression patterns of *LcASC* mRNA upon bacterial infection and LPS and Poly I:C stimulation, the expression levels of *LcASC* were analyzed in the head kidney, liver, and spleen using qPCR. Compared to the control group, the expression levels of *LcASC* mRNA were significantly up-regulated in the head kidney, liver, and spleen after immune stimulation, especially upon bacterial infection. In the head kidney (Figure 3A), the transcription of *LcASC* was significantly up-regulated at 12 h post-injection with LPS and then returned to the control level. Poly I:C stimulation induced a sustainable high expression of *LcASC* and reached a peak of 27.46-fold compared to the control at 48 h. After the *V. parahaemolyticus* challenge, the *LcASC* transcription level immediately increased up to the maximum level at 3 h (85.13-fold) compared to the control and then returned to the basal level. In the liver (Figure 3B), *LcASC* mRNA expression increased at 6 h, 12 h, and 72 h post-injection with LPS. Similarly, significant up-regulation caused by Poly I:C was found at 3 h (8.81-fold), 6 h (7.04-fold,) and 72 h (8.18-fold). In the *V. parahaemolyticus*-infected group, the transcripts were drastically up-regulated and reached a peak (157.17-fold) at 6 h (*p* < 0.05), whereas they returned to the normal level from 12 h to 48 h. Finally, the expression increased at 72 h (4.53-fold) (*p* < 0.05). In the spleen (Figure 3C), when the fish were injected with LPS, significant inductions were noticed at 6 h (2.58-fold), 12 h (3.46-fold), and 48 h (1.53-fold). For the Poly I:C group, *LcASC* mRNA maintained a stable and sustainable increased expression level (1.72–3.96 folds) during the whole stimulation process. Post-challenge with *V. parahaemolyticus*, the significant up-regulation of *LcASC* transcripts emerged at 3 h (4.0-fold), 6 h (22.40-fold), and 12 h (2.72-fold) (*p* < 0.05). In addition, PBS injection had no obvious effects on the expression of *LcASC* in the liver, spleen, and head kidney.

### 2.4. Subcellular Localization of LcASC Protein in Large Yellow Kidney Cells

First, the subcellular localization of LcASC was predicted to be in the cytoplasm by the Softberry website. To better understand the functions of the protein, we further determined the subcellular distribution of pEGFP-N1-LcASC in large yellow kidney cells. The results showed that the empty vector pEGFP-N1 in the control group was uniformly distributed in the whole cell, while the green fluorescence of pEGFP-N1-LcASC in the experimental group was primarily distributed in the cytoplasm and formed a green speck close to the nucleus (Figure 4A). Meanwhile, the exogenous LcASC protein was confirmed to be expressed in cells by Western blot using the anti-GFP antibody, which yielded a specific band with the expected molecular mass (~47.86 kDa) in pEGFP-N1-LcASC-transfected cells but not in empty-vector-transfected cells (Figure 4B).

### 2.5. Transcriptome Analysis after LcASC Overexpression in HEK 293T Cells

The change in protein expression was detected by Western blot after LcASC overexpression in HEK 293T cells. The results showed the protein was significantly increased in LcASC overexpression cells compared to the control empty-vector-transfected cells (Appendix A).

For PCA analysis (Appendix A), the ASC-treated group (three biological replicates) and the empty vector control group (three biological replicates) were clearly separated, which showed that the transcriptome data are reliable. A total of 46 differently expressed genes (DEGs), including 44 up-regulated and 2 down-regulated genes, were checked in a comparison between the overexpression group and the control group (Figure 5, Appendix A). It is worth noting that the transcripts of *FOS*, *KIAA0754*, *POMK*, *NCR3LG1*, *FOS*, *TSIX*, *SLC2A3*, and *LBH* genes were significantly up-regulated in LcASC overexpression cells compared to the control. In contrast, *DOK3* and *RNA28SN5* were emphatically down-regulated (Table 1). Regarding the other two inflammasome components, the NLRP family protein and pro-casepase-1, a difference in transcripts was not detected.

### 2.6. GO and KEGG Enrichment Analysis of DEGs

The 46 DEGs screened were separated into three main categories: Biological processes, cellular components, and molecular functions (Figure 6). Each process was comprised of 10 functional classes. For the biological process category, the regulation of cell growth, gland development, and cell growth was dominant. For the cellular component, the apical part of the cell was identified as the most abundant. Among molecular functions, glycosaminoglycan binding, sulfur compound binding, and microtubule binding accounted for the major proportion.

Pathway-based analysis was an alternative approach for the functional categorization of DEGs. KEGG analysis showed that the DEGs were primarily involved in five metabolic pathways, including parathyroid hormone synthesis, secretion and action, endocrine resistance, salmonella infection, ferroptosis, and cancer signal pathways (Figure 7). The pathways with the most genes were cancer (*FOS*, *EGFR, NOTCH2*, *KIF5C*, *HMOX1*; 5, 22.73%), salmonella infection (*FOS*, *AHNAK*, *KIF5C*, *AHNAK2*; 4, 18.18%), parathyroid hormone synthesis, secretion, and action (*FOS*, *EGFR*, *EGR1*, *MAFB*; 4, 18.18%), endocrine resistance (*FOS*, *EGFR*, *NOTCH2*; 3, 13.63%), and ferroptosis (*HMOX1*, *SAT1*; 2, 9.09%). Each DEG is involved in different signal pathways, among which EGFR was mapped onto the most signal pathways (Appendix A), and to a lesser extent, FOS.

## 3. Discussion

ASC is a pivotal adaptor protein that contributes to innate immunity through the assembly of inflammasome complexes. Our knowledge of the detailed functions of ASC in immune defense is primarily derived from mammalian studies, and thus, investigations on low vertebrates will improve our understanding of the conserved and divergent roles of ASC across species. In the present work, an ASC gene from a large yellow croaker (named LcASC) was characterized via expression detection and transcriptome analysis on the regulatory pathway.

LcASC contains two functional domains, PYD and CARD, which are important for binding to sensor molecules NLRP PRRs and effector protein Caspases. ASC binds to NLRP3 via PYD–PYD interactions in mammals [32]. ASC and NLRP3 have been found to co-localize in the cytosol in Japanese flounder [22] and zebrafish [24], the deletion of PYD in ASC cannot co-localize with NLRP3, and CARD deletion does not influence co-localization. Upon further pulldown detection, Japanese flounder ASC can interact with NLRP3 but not NLRP3-ΔPYD, indicating that the interactions between ASC and NLRP3 depend on the PYD–PYD interaction in teleost [22]. Human ASC binds to NLRP1 depending on the CARD–CARD interaction even though NLRP1 possesses PYD and CARD domains [33]. Zebrafish ASC also binds to NLRP1 through CARD–CARD because NLRP1 has only a CARD domain as in mice [2]. In the large yellow croaker, ASC might recruit NLRP1 via CARD–CARD because it is the only CARD domain in NLRP1. For the effector protein, in all vertebrates except for zebrafish, pro-Casp1 has two functional domains, CARD and caspase consensus. ASC interacts with pro-Casp1 through CARD–CARD in most vertebrates. However, in zebrafish, ASC binds with pro-Casp1 homolog pro-CaspA and pro-CaspyB, which possess PYD and caspase consensus domains instead of CARD, so the PYD–PYD interaction is its distinguishing feature [2]. In addition, there is a linker region between PYD and CARD. This region is less conserved than the PYD and CARD domains among vertebrates. In human ASC, the linker region contributes to the formation of ASC three-dimensional construction to avoid the binding interface of each domain and has no effects on PYD and CARD interactions.

*LcASC* transcripts were ubiquitously expressed in all examined tissues with dominant expression in the spleen followed by the gills and skin. The spleen is the most important immune organ, and the gills and skin are the first protective barrier in fish against invading pathogens. The expression pattern thus indicated that *LcASC* plays significant immune roles. A similar expression pattern has been observed in goldfish [23] and zebrafish [24] with higher mRNA levels in the spleen and gills. The *ASC* expression in Japanese medaka [28] tissues was higher in the skin and gills. In the orange-spotted grouper [25], *ASC* transcripts were primarily expressed in the head-kidney and gills. The *ASC* gene from mandarin fish was prominently expressed in the head kidney and spleen [21]. Human ASC protein/mRNA levels were primarily expressed in the skin, spleen, and immune cells, particularly neutrophils and monocytes [9]. In addition, *LcASC* mRNA has a slight presence in the liver, brain, kidney, and blood, showing that secretion of LcASC may occur, if anywhere, in limited cells.

In previous studies, several reports have shown that the inflammasome contributes to protection against infectious pathogens. In humans, the expression of *ASC* was induced by LPS stimulation in intestinal mucosa and neutrophils [34]. In turbot, *ASC* expression restricts *E. piscicida* in several organs, including the liver, spleen, kidney, and intestine [26]. In Japanese flounder, the *ASC* was induced in HKM (head kidney macrophages) and PBL (peripheral blood leucocytes) after stimulation with LPS, Poly I:C, and zymosan [22]. In goldfish, a significant induction of *ASC* was observed after nigericin challenge in macrophages [23]. In the orange-spotted grouper, the transcriptional levels of *ASC* gradually increased depending on post-stimulation time [25]. However, in medaka [28] and zebrafish embryos [24], *ASC* gene expression has no obvious changes after *E. piscicida* and *Staphylococcus aureus* infection, respectively. In our studies, *ASC* transcripts piled on after different immune stimulation, especially *V. parahaemolyticus*. *ASC* transcripts sharply increased during the early phase of *V. parahaemolyticus* infection, and in contrast, the virus-like stimulant Poly I:C caused ASC transcripts to increase in a later period. The results might be due to different infection mechanisms between the bacteria and virus, which also indicated that the expression of *ASC* has different responses to various pathogens.

Subcellular localization analysis showed that LcASC exhibited a clear speck-like aggregation in the cytoplasm close to the nucleus, consistent with the previously reported localization of ASC in fish [23,24,26]. A study on zebrafish showed that the formation of spots depends on several residues in the PYD domain [24]. Moreover, Wang et al. and Li et al. reported that the PYD domain of ASC could form filamentous structures when it was overexpressed in zebrafish [24] and turbot cells [26], while fewer filaments were observed in ASC CARD-overexpressing cells.

To further inquire into the potential immune regulatory pathway of LcASC, 46 DGEs were found by transcriptome analysis in LcASC-overexpressing cells versus control. Most of the DGEs were associated with various cancers through KEGG analysis, suggesting that ASC might regulate inflammation and cell growth, differentiation, and apoptosis. GO functional classes also indicated that DGEs might be involved in cell growth. ASC has been approved to regulate inflammation and cell pyroptosis [32,35]. Whether ASC can regulate cell growth and differentiation needs to be clarified in future studies. Importantly, the up-regulated *FOS* gene was involved in many signaling pathways, such as bacterial infection, cancer, programmed cell death, etc. *FOS* is a nucleoprotein encoded by mature mRNA generated from *c-fos* gene transcription. *c-Fos* is an immediate early-response gene that binds the c-Jun protein to form transcription factor active protein1 (AP-1), one of the most powerful transcriptional factors of the immune system [36]. The suppression of *c-fos* expression via siRNA markedly increased *Brucella* survival in macrophages [37]. On the other hand, the down-regulated gene *DOK3*, downstream of the kinase family, functions primarily as an adaptor to facilitate protein–protein interactions since it has no catalytic activity. There is increasing evidence to indicate that it is involved in various immune receptors’ signaling. First, it restricts the Ca^2+^ and JNK (Jun N-terminal kinase) activation in B cell receptor signaling [38,39,40]. Secondly, DOK3 negatively regulates TLR4 (Toll-like receptor 4) signaling by limiting LPS-induced ERK (extracellular regulated protein kinases) activation and cytokine responses in macrophages [39]. Finally, Dok-3 plays a critical and positive role in TLR3 signaling by enabling TRAF3 (TNF receptor associated factor 3)/TBK1 (TANK Binding Kinase 1) complex formation and facilitating TBK1 and IFN (Interferon) regulatory factor 3 activation and the induction of IFN-β production [40]. The results of the present study raise the question of whether there is another signaling pathway in addition to NLR-ASC-Caspase in large yellow croaker. Future studies should primarily focus on the identification of the upstream and downstream molecular signals of the *ASC* gene and explore the ASC-mediated pyroptosis process in the large yellow croaker. Finally, a hypothesis of ASC’s role in most fish is illustrated in Figure 8.

## 4. Material and Methods

### 4.1. Ethical Statement

All experiments were carried out following the principles and protocols of the Animal Care and Use Committee at Fisheries College, Jimei University, China. The experimental fish were anesthetized with tricaine methane sulfonate (MS-222, Sigma, St. Louis, MO, USA) for sample collection and anatomical observation.

### 4.2. Fish and Sample

Health large yellow croakers weighing 125–135 g were obtained from Guanjingyang company in Ningde city, Fujian province, China. The fish were acclimatized in filtered and aerated seawater (28 °C) and fed with commercial feed twice daily for two weeks before conducting the experiments.

For cDNA clone and tissue distribution analysis, five healthy fish were dissected and a series of tissues including skin, gill, muscle, liver, spleen, intestine, kidney, stomach, heart, head kidney, and brain tissues were isolated. Blood was taken from the fishtail vein. Blood cells were separated immediately via centrifugation at 3000× *g* for 10 min at 4 °C. All samples were snap-frozen immediately in liquid nitrogen and stored at −80 °C until use.

For the challenge experiment, 240 healthy fish were randomly divided into four groups, including three test groups and one control group. Netx, 250 μL of the immune stimulants LPS (1 mg mL^−1^), poly I:C (1 mg mL^−1^), and *V. parahaemolyticus* (1 × 10^6^ CFU μL^−1^) were injected intraperitoneally into the fish in three test groups, respectively. Meanwhile, the control group was injected with 250 μL PBS and treated using the same procedures as those performed in the experiment group. At seven different time points (0, 3, 6, 12, 24, 48, and 72 h) post-infection, five individuals were randomly selected from the control group and experimental groups. The samples of the head kidney, liver, and spleen were isolated and preserved in liquid nitrogen before total RNA extraction.

### 4.3. RNA Extraction and cDNA Synthesis

Total RNA from the collected tissue samples of the large yellow croaker was isolated using the TRIzol reagent (Invitrogen, Waltham, MA, USA) following the manufacturer’s protocols. The RNA was incubated with RNase-free DNaseI (Promega, Madison, WI, USA) at 37 °C for 30 min to remove the DNA. The quality of the isolated RNA was evaluated via 1.5% agarose gel electrophoresis, and the RNA concentration was measured using a NanoDrop 2000 Spectrophotometer (Thermo Scientific, Waltham, MA, USA).

First-strand cDNA was synthesized in a 20 μL reaction mixture containing a pool of RNA 2 μg using the PrimeScript™ II 1st strand cDNA Synthesis Kit (Takara, Higashiosaka, Japan). The synthesized cDNA was diluted 100-fold in nuclease-free water and stored at −20 °C until further use.

### 4.4. Identification of cDNA and Sequence Analysis

Through splicing and comparison, the full-length cDNA sequence of *LcASC* was obtained from the *L. crocea* transcriptome database constructed in our laboratory. A pair of primers (cASC-F and cASC–R) (Appendix A) and spleen cDNA were used in PCR to amplify the whole *LcASC* ORF sequence. PCR was conducted using a Thermal Cycler 2720 (Thermo fisher) under the following amplification conditions: 94 °C for 5 min, followed by 35 cycles at 94 °C for 30 s, 55 °C for 30 s, and 72 °C for 2 min, with a final extension at 72 °C for 10 min.

The cDNA and amino acid sequences of LcASC were analyzed using the BLAST program at the National Center for Biotechnology Information (NCBI). The molecular weight (MW) and theoretical isoelectric point (pI) of LcASC were predicted using EXPASY (http://www.expasy.org/) (28 November 2022). Phosphorylation prediction was performed using the NetPhos 2.0 Server (http://www.cbs.dtu.dk/services/NetPhos/) (28 November 2022). Multiple sequence alignment was performed using the deduced amino acid sequences of LcASC and those of other closely related species from BLAST. The putative domain of LcASC was analyzed by the SMART program (http://smart.embl-heidelberg.de/) (28 November 2022). PSIPRED 4.0 (http://bioinf.cs.ucl.ac.uk/psipred) (29 November 2022) and SWISS-MODEL (http://www.swissmodel.expasy.org/) (29 November 2022) were applied to predict the secondary and tertiary structures of the protein. The phylogenetic tree was constructed based on the amino acid sequences of vertebrate ASC proteins using the maximum likelihood method with ClustalX and MEGA 10.2.6 software.

### 4.5. Tissue Distribution of LcASC and Expression Pattern Analysis

To detect gene expression, qPCR was performed using the StepOnePlus Detection System (ABI, Foster City, CA, USA) and ChamQ Universal SYBR qPCR Master Mix Kit (Vazyme, Nanjing, China) following the user’s manual. The specific primers of qASC-F/R used for qPCR are listed in Appendix A. All samples were amplified in triplicate, and βactin was used as an internal control. The PCR conditions were as follows: 95 °C for 15 min; 40 cycles of 95 °C for 10 s; 60 °C for 20 s; and 72 °C for 32 s. Relative expression was calculated by the 2^−ΔΔCt^ method. The results were expressed as mean ± SEM, and statistical analysis was performed using SPSS 20.0 software. *p* < 0.05 was considered statistically significant.

### 4.6. Subcellular Localization and Overexpression Analysis

The subcellular localization analysis was carried out as described in previous work [25]. In brief, the full sequence of LcASC ORF was amplified with specific primers for sASC-F/R, as shown in Appendix A. The whole CDS region of LcASC was inserted into the pEGFP-N1 plasmid to construct a eukaryotic expression vector. pEGFP-N1-LcASC and pEGFP-N1 were extracted with a small amount of the plasmid extraction kit (TIANMO, China). Large yellow croaker kidney cell lines were inoculated onto a 24-well cell culture plate and cultured until cells adhered to 80–90%. The pEGFP-N1-LcASC and pEGFP-N1 were transfected into the kidney cells for 24 h by the electron-transfected method, and the cells were then fixed and stained. Finally, the cells were observed with laser scanning confocal observation (Leica TCS SP8 system, Berlin, Germany) under two channels of DAPI excitation light and GFP excitation light.

For overexpression, the whole CDS region of LcASC was inserted into the pcDNA3.1 vector and transfected into HEK 293T cells using the Lipo8000 TM transfection reagent. After 24 h post-transfection, the cells were harvested for Western blot to examine the expression level of the LcASC protein.

### 4.7. Western Blot Analysis

SDS-PAGE and assays were performed using routine methods. The cells were treated in RIPA lysis buffer, whereafter the lysed cell protein was centrifugated. The separated proteins were analyzed through 15% SDS-PAGE and then transferred onto a polyvinylidene difluoride (PVDF) membrane (Millipore, Darmstadt, Germany). The membrane was blocked with 2% BSA (Vazyme, China) and incubated with the rabbit anti-His antibody (1:1000, ABclonal, Wuhan, China). The target proteins were visualized with chemical luminescence substrate with Amersham Imager 600 (GE, Atlanta, GA, USA) after incubation with HRP conjugated secondary antibodies goat anti-rabbit (ABclonal, China, 1:2000 dilution).

### 4.8. Library Construction and Detection

RNA was isolated from the control and treatment samples (3 biological replicates each) using the Trizol reagent (Invitrogen, USA). The extracted RNA was determined using electrophoresis. After quality control, mRNA was subjected to the enrichment of eukaryotic RNA using oligo (dT) beads. The cDNA library was constructed following the manufacturer’s protocol (Illumina, San Deigo, CA, USA), and Qubit2.0 was then used for preliminary quantification. Agilent 2100 was used to check the libraries. After satisfying the requirements, the libraries quantified the effective concentrations (the effective concentrations of the libraries were greater than 2 nmol L^−1^) via qPCR to ensure that they were of good quality. Libraries were sequenced by the Illumina HiSeq6000 platform.

### 4.9. Transcript Assembly and Differently Expressed Genes (DGEs) Analysis

Raw reads of the RNA-seq were subjected to filtration to generate clean reads using FastQC software. The clean data were subsequently aligned to the human reference genome (GenBank: GRCH38) by STAR software. Transcriptome assembly for all clean data was accomplished using StringTie software.

For DGEs analysis, transcripts were quantified using FeatureCounts to calculate the transcript abundance. The principal component analysis (PCA) was conducted to visualize the relative differences in transcriptional profile between the pcDNA3.1-ASC transfection group and the empty vector control. DEGs were identified using DEseq2 based on the criteria of |log2FC| ≥ 0.5, *p* ≤ 0.01.

### 4.10. Enrichment Analysis of DEGs

GO (Gene Ontology, http://www.geneontology.org/) enrichment analysis of differently expressed genes (DEGs) was conducted with GOseq based on Wallenius non-central hyper-geometric distribution. KEGG (Kyoto Encyclopedia of Genes and Genomes, http://www.genome.jp/kegg/) pathway enrichment analysis of DEGs was conducted via clusterProfiler in R package.

## 5. Conclusions

In summary, our study describes an *ASC* gene in large yellow croaker, namely, *LcASC*. The gene conserves the PYD and CARD domains, which are responsible for assembling inflammasomes. The *LcASC* gene was notably expressed in the spleen, gills, and skin. The transcripts of *LcASC* mRNA were dramatically induced after immune stimulation, especially bacterial infection. LcASC was localized in the cytoplasm and formed a focused speck close to the nucleus. Furthermore, we found 46 DEGs in a comparative transcriptome analysis between the LcASC overexpression and control vector groups. Noticeably, the up-regulated gene *Fos* and down-regulated *DOK3* gene might be involved in important immune-regulated roles in the ASC signal pathway.

## Figures and Tables

**Figure 1 ijms-24-02175-f001:**
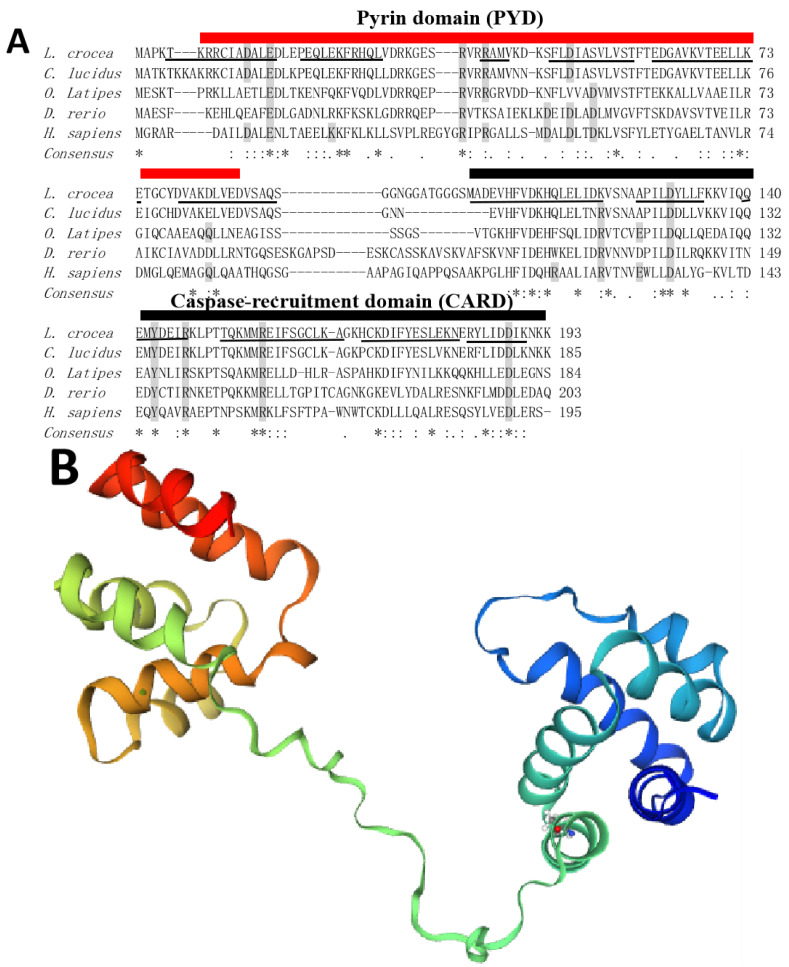
(**A**) Multiple sequence alignment of ASC from different species. Gray shading denotes some charged residues for mammalian PYD or CARD fibrillation. The underline represents α-helix structure. (**B**) Tertiary structure of LcASC protein containing twelve α-helix structures. (**C**) The phylogenetic tree of LcASC.

**Figure 2 ijms-24-02175-f002:**
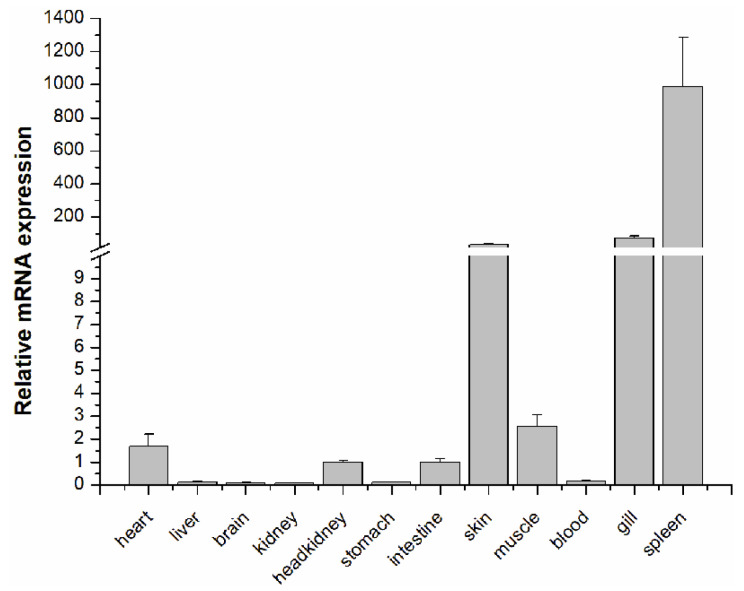
Tissues distribution of *LcASC*. The relative expression profile of *LcASC* was examined by qPCR and normalized against β-actin. Replications (*n* = 5) were performed in each isoform and displayed as mean ± S.D.

**Figure 3 ijms-24-02175-f003:**
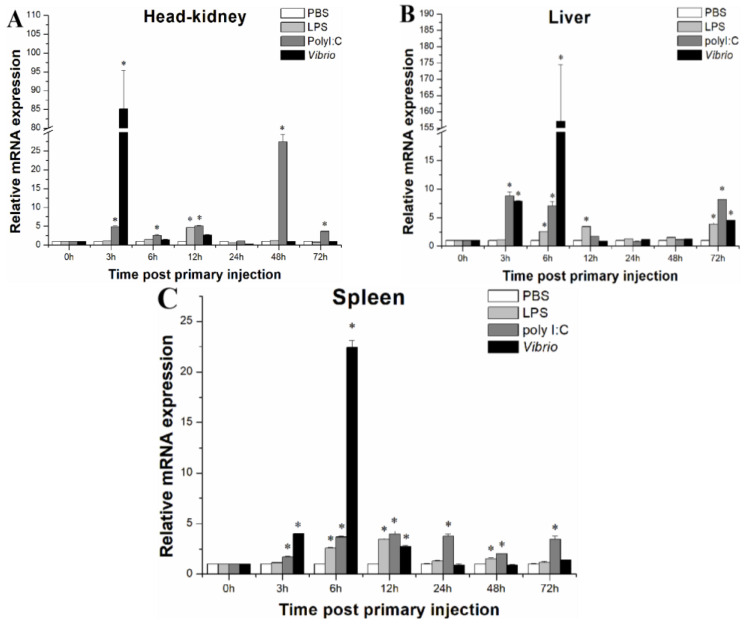
The expression level of *LcASC* upon various immune stimulation in large yellow croaker head kidney (**A**), liver (**B**), and spleen (**C**). The expression levels of *LcASC* were examined by qPCR. Data are expressed as the relative expression value (means ± S.D., *n* = 5) and normalized against β-actin (* *p* < 0.05). The experiment was conducted in three replications.

**Figure 4 ijms-24-02175-f004:**
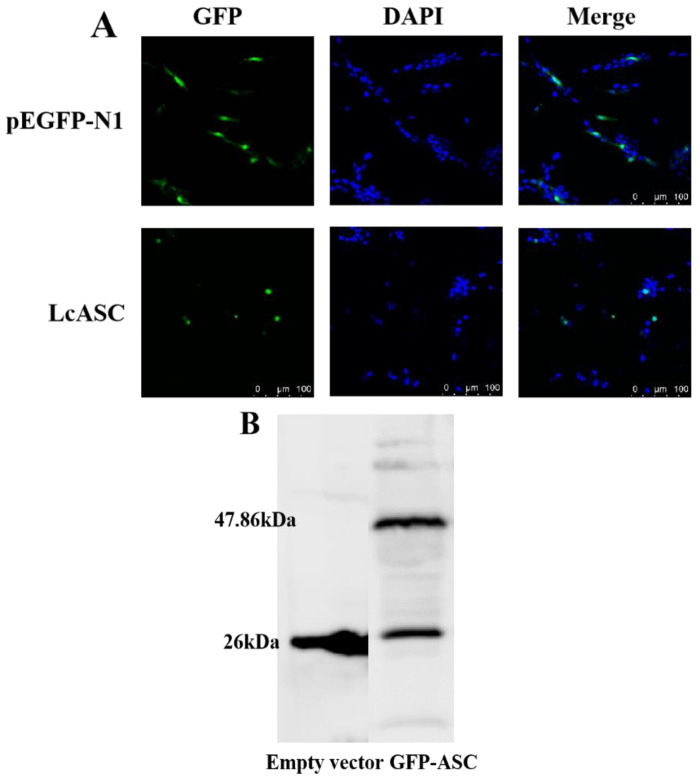
(**A**) Cytoplasmic localization of LcASC protein in large yellow croaker kidney cells. (**B**) The exogenous expression of LcASC protein in large yellow croaker kidney cells confirmed by Western blot using anti-GFP antibody. The cells transfected with GFP-tagged LcASC plasmid show a ~47.86 kDa band, which is absent in control cells transfected with empty vector.

**Figure 5 ijms-24-02175-f005:**
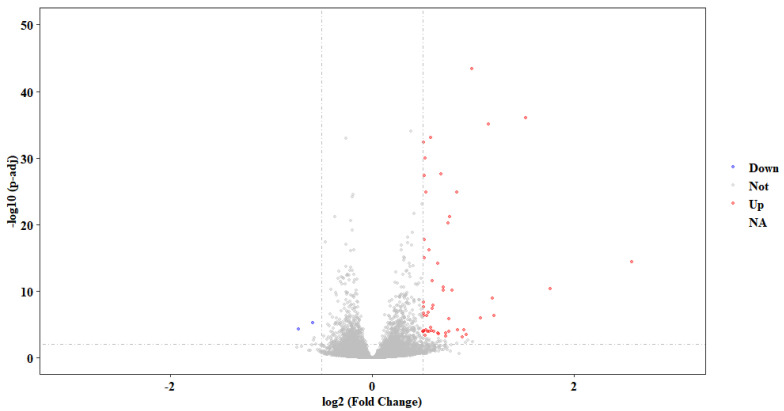
The volcano plot of differently expressed genes in LcASC overexpression samples. The red dots indicate the up-regulated genes, the blue dots indicate the down-regulated genes, and the grey dots indicate unchanged genes.

**Figure 6 ijms-24-02175-f006:**
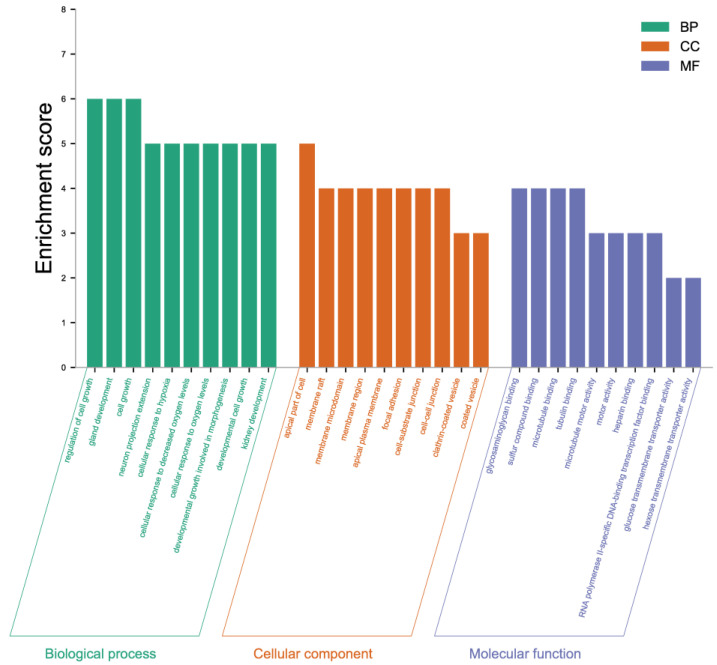
Column chart of GO enrichment analysis of differently expressed genes. The screened DEGs were separated into three main categories: Biological processes, cellular components, and molecular functions.

**Figure 7 ijms-24-02175-f007:**
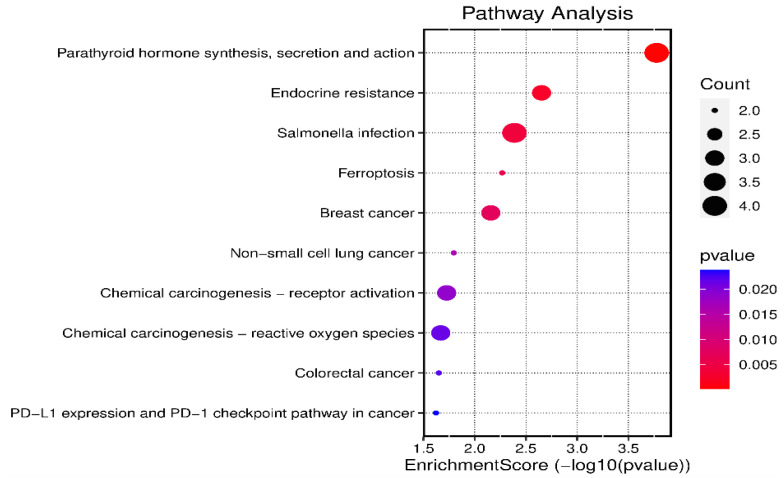
Bubble chart of GO enrichment analysis of differently expressed genes. The size of bubble represents the counts of DGEs, and the different colors represent the significance of the difference (*p* value).

**Figure 8 ijms-24-02175-f008:**
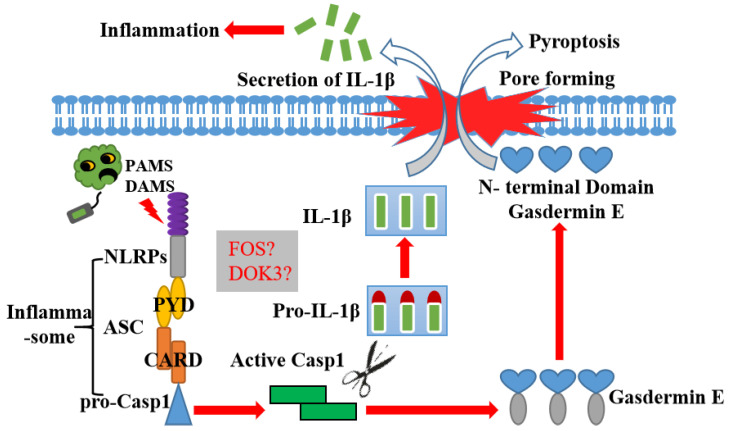
Putative inflammasome signal pathway in most fish.

**Table 1 ijms-24-02175-t001:** The function annotation of several important DGEs.

GeneID	log_2_^FoldChange^	Function Annotation
KIAA0754	2.575797	Related to autoimmune syndrome, Sjögren’s syndrome, metabolic syndrome, and inflammation
POMK	1.761321	Regulates dystroglycan function, as candidate gene for Walker-Warburg syndrome with meningoencephalocele and muscular dystrophy-dystroglycanopathy
NCR3LG1	1.522483	Regulates tumor cell response
FOS	1.21052	Binds c-Jun protein to form transcription factor active protein1
TSIX	1.1931	participates in the choice of the inactive X and in Xist regulation
SLC2A3	1.153975	acts as a tumor promoter
LBH	1.077569	Related to cancer and Rheumatoid Arthritis
DOK3	−0.58951	Regulates B cell receptor, TLR3 and TLR4 signaling
RNA28SN5	−0.72932	uncharacterized

## Data Availability

Data will be made available upon request.

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
