# Peer review of "Molecular Characterization, Expression, and Regulatory Signal Pathway Analysis of Inflammasome Component Apoptosis-Associated Speck-like Protein Containing a CARD Domain (ASC) in Large Yellow Croaker (Larimichthys crocea)"

_ijms, 2023, doi:10.3390/ijms24032175_

Round 1

Reviewer 1 Report

The study have described the molecular characterization, expression profile, localization, and predicted function of LcASC in large yellow croaker. In addition, the up-regulated gene Fos and down-regulated DOK3 gene might be involved in important immune regulated roles in ASC pathway. The content of manuscript is logical, rich and clear. While this manuscript has many obvious mistakes, it is recommended to check carefully before submitting.

-Abstract The dot of “Vibrio. Parahaemolyticus” should be deleted?

-There is no line number

-The figure legends of Figure 5, 6 and 7 should be detailed

-The names of species should be italic

Author Response

Reviewer 1: The study have described the molecular characterization, expression profile, localization, and predicted function of LcASC in large yellow croaker. In addition, the up-regulated gene Fos and down-regulated DOK3 gene might be involved in important immune regulated roles in ASC pathway. The content of manuscript is logical, rich and clear. While this manuscript has many obvious mistakes, it is recommended to check carefully before submitting.

1. Abstract The dot of “Vibrio. Parahaemolyticus” should be deleted?

Re:Thanks for your review. We have revised the “Vibrio. Parahaemolyticus” into Vibrio Parahaemolyticus.

2. There is no line number.

Re:Thank for very much. We have added the line number.

3. The figure legends of Figure 5, 6 and 7 should be detailed. 

Re:Thanks for your advice. We have rewritten the legends of Figure 5, 6 and 7 in details.

4. The names of species should be italic.

Re:Sorry, we neglected the italic when standardizing the paper format. Now, we have revised it carefully.

Reviewer 2 Report

This study focused on ACS, an important immune gene of the inflammasome component in a commercial fish species, large yellow croaker, and identified its immunoreactivity in vivo based on the tissue distribution and immune response. Furthermore, the regulatory expressional network of this gene was revealed through the DEGs and enriched pathways. However, the logic of writing is not clear enough, and the presentation of the data was not sufficient. Therefore, a moderate revision is suggested.

Introduction section:

1.      How is ASC involved in the inflammasome complex in mammalia, structurally and functionally? This should be introduced briefly.

2.      The introduction of fish ASC was also limited yet. More references can be consulted to provide more details about fish ASC's previous research, e.g., structure-related studies can be included in the third paragraph.

3.      There lacks an introduction for currently used immunostimulants in the background of fish inflammasome-related studies.

4.      How is the inflammasome involved in the fish disease? The background for the inflammasome related studies in large yellow croaker has not been mentioned.

5.      The purpose of this study has not been pointed out.

Results section:

6.      The important DEGs could be listed as a separate table to show more annotation details.

7.      Did the author deposit the transcriptomic data in a publicly available database? It is necessary for other researchers to verify this study or reuse the data.

Discussion section:

8.      The discussion is not in a clear logic.

9.      According to the result from different immunostimulants, the meaning of different responses in different time points and tissues need be explained. Since the reactions are different to virus-like stimulant and bacterium-like stimulant, the involved mechanism is worthy of a discussion.

10.   The enriched GO terms and KEGG pathways were not discussed sufficiently. The mentioned genes were not found in the result section. It is better to show important DEGs in a table.

11.   Based on the sufficient discussion, a hypothesis of ASC's role in fish can be illustrated as a new diagram.

Other minor problems:

1.      The first letter of "Western blot" should be capitalized.

2.      All scientific names should be italicized.

3.      A space should be placed after the period in name: "L.crocea"

4.      There are some misuse of Chinese punctuation in the text.

Author Response

Reviewer 2: This study focused on ACS, an important immune gene of the inflammasome component in a commercial fish species, large yellow croaker, and identified its immunoreactivity in vivo based on the tissue distribution and immune response. Furthermore, the regulatory expressional network of this gene was revealed through the DEGs and enriched pathways. However, the logic of writing is not clear enough, and the presentation of the data was not sufficient. Therefore, a moderate revision is suggested. 

Introduction section:

1.How is ASC involved in the inflammasome complex in mammalia, structurally and functionally? This should be introduced briefly.

Re:Thanks for your review. We have added the introduction on mammal ASC involved the inflammasome structure and function in line 45-48.  

2. The introduction of fish ASC was also limited yet. More references can be consulted to provide more details about fish ASC's previous research, e.g., structure-related studies can be included in the third paragraph.

Re:Thank for very much. The structure-related studies on fish ASC have been provided at length in discussion paragraph 2 line 189-204.

3.There lacks an introduction for currently used immunostimulants in the background of fish inflammasome-related studies.

Re:Thank for very much. We have added immunostimulants background in introduction part in line 60-62, and further discussed in discussion paragraph 4 line 214-226. 

4. How is the inflammasome involved in the fish disease? The background for the inflammasome related studies in large yellow croaker has not been mentioned.

Re:Thank for very much. In fish, how is the inflammasome involved in the fish disease remaining confused. We have added the background for the inflammasome related studies in large yellow croaker in line 67-69.

5. The purpose of this study has not been pointed out.

Re:Thank for your advice. The purpose of this study is to explore ASC immune mechanism in large yellow croaker, then further to investigate inflammasome’ roles. We have added it in line 70-71.

Results section:

1. The important DEGs could be listed as a separate table to show more annotation details.

Re: Thank you for your good advice. We have added a separate table 1 in result line 162.

2. Did the author deposit the transcriptomic data in a publicly available database? It is necessary for other researchers to verify this study or reuse the data.

Re:Thank you very much. We have submitted the transcriptomic data in NCBI.

Discussion section:

The discussion is not in a clear logic.

1. According to the result from different immunostimulants, the meaning of different responses in different time points and tissues need be explained. Since the reactions are different to virus-like stimulant and bacterium-like stimulant, the involved mechanism is worthy of a discussion.

Re:Thank you for your good advice. The immune responses are very complicated, thus it is very difficult to discuss in details, and we found that others also did not discussed that for other genes. We have rewritten that in line 222-226. 

2. The enriched GO terms and KEGG pathways were not discussed sufficiently. The mentioned genes were not found in the result section. It is better to show important DEGs in a table.

Re:Thank you very much. We have added the discussion on the enriched GO terms and KEGG pathways in line 233-237. Moreover, we have added a separate table 1 for some important DEGs in result line 162.

3. Based on the sufficient discussion, a hypothesis of ASC's role in fish can be illustrated as a new diagram.

Re:Thank you for your good advice. We have drawn a new diagram to illustrate a hypothesis of ASC's role in the most of fish in Fig.8 line 254.

Other minor problems:

1.The first letter of "Western blot" should be capitalized.

Re: Thank you very much. We have revised it in the whole paper.

2. All scientific names should be italicized.

Re:Sorry, we neglected the italic when standardizing the paper format. Now, we have revised it carefully.

3. A space should be placed after the period in name: "L.crocea"

Re:Thank you very much. We have added a space in name "L.crocea".

4. There are some misuse of Chinese punctuation in the text.

Re:Thank you very much. We have carefully revised it in the whole paper.

Reviewer 3 Report

In the article "Molecular characterization, expression and regulatory signal pathway analysis of inflammasome component apoptosis-associated speck-like protein containing a CARD domain (ASC) in large yellow croaker (Larimichthys crocea)" the authors describe the ASC gene in large yellow croaker, mRNA expression in different internal organs.

The article is written in an understandable way. The introduction provides all the necessary information to understand the main topic of the article.

The material and methods part describes in detail preparation to the experiment and applied methods.

All tables and figures are readable and put in the right place in the text.

Results are presented very clearly and understandably.

The discussion refers to other references and refers to obtained results.

Although these positive comments I have several suggestions:

-          All Latin names should be written in italic (even in the title), check carefully all text

-          There is unnecessary space in the title - correct it

-          In the introduction part - explain WT shortcut

-          Results. 2.1. - the first, the fifth line -unnecessary space - appears several times in the text, check carefully all text

-          qRT-PCR? - Do you mean Real-Time Quantitative Reverse Transcription PCR -Real Time RT-PCR (qRT-PCR) or Real-Time Quantitative PCR - Real Time PCR (qPCR) ??? because the description fits to Real Time PCR (qPCR), not qRT-PCR - it appears several times in the text and in the description of Figure 3

-          2.6 title - move the title to next page

-          In the discussion part explain shortcuts: HKM, ERK, TLR - appear the first time in the text without explanation

-          4.2 part - use superscript: 1x106 or µl-1

-          Page 10 of 16 - the third line - new sentence after the dot should be a big letter.

-          Page 10 of 16 - the 10th line - head kidney - a small letter

-          4.5 part - β-actin primers- nucleotide sequences should also appear in the Table S1

-          4.8 part - use superscript, correct qRT-PCR

-          References should be described as follows:

Journal Articles:

Author 1, A.B.; Author 2, C.D. Title of the article. Abbreviated Journal Name Year, Volume, page range.

-          Table S3- space between Table and S3. Columns p.adjust and q value are connected - separate them

Author Response

Reviewer 3: In the article "Molecular characterization, expression and regulatory signal pathway analysis of inflammasome component apoptosis-associated speck-like protein containing a CARD domain (ASC) in large yellow croaker (Larimichthys crocea)" the authors describe the ASC gene in large yellow croaker, mRNA expression in different internal organs.

The article is written in an understandable way. The introduction provides all the necessary information to understand the main topic of the article.

The material and methods part describes in detail preparation to the experiment and applied methods.

All tables and figures are readable and put in the right place in the text.

Results are presented very clearly and understandably.

The discussion refers to other references and refers to obtained results.

Although these positive comments I have several suggestions:

1. All Latin names should be written in italic (even in the title), check carefully all text.

Re:Sorry, we neglected the italic when standardizing the paper format. Now, we have revised it carefully.

2. There is unnecessary space in the title - correct it.

Re:Thank you very much. We have deleted the space.

3. In the introduction part - explain WT shortcut.

Re:Thank you very much. We have revised it in line 51-52.

4. 2.1. - the first, the fifth line -unnecessary space - appears several times in the text, check carefully all text.

Re:Thank you very much. We have corrected it.

5. qRT-PCR? - Do you mean Real-Time Quantitative Reverse Transcription PCR -Real Time RT-PCR (qRT-PCR) or Real-Time Quantitative PCR - Real Time PCR (qPCR) ??? because the description fits to Real Time PCR (qPCR), not qRT-PCR - it appears several times in the text and in the description of Figure 3.

Re:Thank you very much. We refer to Real-Time Quantitative PCR, and it has been marked at the first occurrence in line 101-102.

6. 2.6 title - move the title to next page.

Re:Thank you very much. We have moved it.

7. In the discussion part explain shortcuts: HKM, ERK, TLR - appear the first time in the text without explanation.

Re:Thank you very much. We have explained them in line 217 and 244-248.

8. 2 part - use superscript: 1x106 or µl-1.

Re:Thank you very much. We have corrected them in line 269-270.

9. Page 9 of 16 - the third line - new sentence after the dot should be a big letter.

Re:Thank you very much. We have corrected it in line 209.

10. Page 10 of 16 - the 13th line - head kidney - a small letter.

Re:Thank you very much. We have corrected it in line 273.

11. 5 part - β-actin primers- nucleotide sequences should also appear in the Table S1.

Re:Thank you very much. We have added it in table S1.

12. 8 part - use superscript, correct qRT-PCR.

Re:Thank you very much. We checked the whole manuscript and corrected qRT-PCR into Real-Time Quantitative PCR (qPCR).

13. References should be described as follows:

Journal Articles:

Author 1, A.B.; Author 2, C.D. Title of the article. Abbreviated Journal Name YearVolume, page range.

Table S3- space between Table and S3. Columns p.adjust and q value are connected - separate them

Re:Thank you very much. We have revised all references.